# The Impacts of Volcanic Activity on Microbial Growth—A Simulation Experiment in the Qiliao Section in Shizhu County, Chongqing, China

**DOI:** 10.3390/biology13110895

**Published:** 2024-11-03

**Authors:** Chaoyong Wang, Qianjin Che, Bin Luo, Yuxuan Zhu, Jie Liu, Mengmeng Tang

**Affiliations:** 1Key Laboratory of Coalbed Methane Resources and Reservoir Formation Process, Ministry of Education, China University of Mining and Technology, Xuzhou 221008, China; wangcy@cumt.edu.cn (C.W.); zhuyuxuan@cumt.edu.cn (Y.Z.); ts23010024a31@cumt.edu.cn (J.L.); tangmengmeng@cumt.edu.cn (M.T.); 2School of Resources and Geosciences, China University of Mining and Technology, Xuzhou 221116, China; 3College of Environment Science and Engineering, Taiyuan University of Technology, Taiyuan 03002, China; luobin01@tyut.edu.cn

**Keywords:** volcano, microbial activity, chlorophyll, stimulation test

## Abstract

This study indicates that volcanic activity has a long-term impact on biological growth. The synthesis of chlorophyll a by *Anabaena pseudoichthyoides* was most efficient in the presence of increasing concentrations of volcanic ash leachate. Analysis of the major and trace elements in the solution before and after cultivation shows that the volcanic ash soaking solution has a higher nutrient content compared to granite. This increased nutrient content is a key factor promoting microbial growth. The findings suggest that volcanic ash significantly influences microorganisms, with lasting effects that can persist for tens of thousands to hundreds of thousands of years.

## 1. Introduction

Black shale, with rich siliceous minerals and organic matter, is deposited in the Wufeng Formation of Late Ordovician and the Longmaxi Formation of Early Silurian in South China [1,2]. These two sets of organic-rich shales developed multilayered potassium-rich porphyritic layers, indicating that volcanism occurred frequently during the Late Ordovician–Early Silurian [3,4,5,6]. In recent years, it has been one of the important research hotspots for the genetic relationship between volcanic eruptions and organic matter enrichment internationally. A set of organic-rich shales (internationally known as “hot shale”) were deposited in North America, Europe, North Africa, and southern China and is indicative of one of the most important source rock formations for Paleozoic oil and gas resources worldwide during this major period [7,8,9,10].

The volcanic ash produced by volcanic eruptions is rich in nutrients, promoting biological development and reproduction and increasing the productivity of ancient oceans [11]. In addition, it has been proposed that a moderate frequency of volcanic activity will have a positive impact on the formation of high-quality marine shale rich in carbon and high in silicon [12]. A large number of studies have shown that volcanic ash in modern volcanic eruptions can release a large amount of nutrients, such as Fe, P, N, Si, Mn, etc., which can dissolve in seawater and promote improvement in plankton productivity in the surface layer of seawater [13,14,15]. For example, the relatively low content of Fe in the Pacific Ocean limited plankton production and blooms. In these regions, the increase in Fe on the molar scale could trigger large-scale diatom blooms, and previous studies have shown that modern volcanic eruptions were closely related to biological blooms in the ocean. It is reported that the 2003 eruption of the Anatahan volcano in the Mariana Islands triggered a large increase in productivity in the depleted trophic zone of the northwestern Pacific region [13]; the 2008 eruption of the Kasatochi volcano in Alaska, USA, triggered a bloom of plankton in the northeast of the Pacific [14,15]. Similarly, the 2010 eruption of Eyjafjallajokulla in Iceland increased biological productivity in the Icelandic basin in the North Atlantic [16,17,18,19,20].

A large number of porphyritic rocks have been found in many reported outcrops and wells in the Sichuan Basin, especially confirming the frequent volcanic activity in the P. pacificus zone of the Late Kaitian stage and the A. ascensus zone of the Early Rudan stage in the Qiliao outcrop [5,6]. However, there is a lack of strong evidence in geological history that volcanic activity impacted surface productivity in the ocean currently; it is difficult to determine because of a lack of discriminative indicators for the duration, frequency, and intensity of volcanic activity in sedimentary strata. The threshold intensity and duration of volcanic activity can be solved through numerical simulation, long-term indoor experiments, and other means. This article attempts to explore this issue through a detailed description of the Qiliao section in Shizhu County and indoor experiments. By adding different concentrations of volcanic ash to the experimental system containing *Pseudonitzschia*, we investigated the impact of volcanic activity on organic matter enrichment. This further explores the distribution patterns and formation mechanisms of organic-rich shales and provides predictions and assessments for optimizing exploration areas for shale gas.

## 2. Materials and Methods

### 2.1. Sample Collection and Determination of Organic Carbon Content

In order to study the effects of volcanic ash on microbial production and organic matter enrichment, samples of 1.3 m intervals (6 layers of volcanic ash and 7 layers of shale) were collected from the Qiliao outcrop in Shizhu County, Chongqing (Figure 1). The total organic carbon content of the samples was determined using an elemental analyzer.

### 2.2. Experimental Design

The samples of volcanic ash and granite were ground into powder, a column was filled with 151 g of the powder, and then the column was soaked in saltwater. The leachate was then obtained by filtration using 0.45 µm filtering membranes. Seven different leachate concentrations were set in this experiment as follows: 1/10, 1/100, 1/500, 1/1000, 1/1500, 1/2000, and 1/2500.

### 2.3. Bacterial Culture and Inoculation

The *Anabaena pseudoichthyoides* (EACHB-82) that had been used in this experiment were purchased from the Freshwater Algae Species Bank (FACHB) of the Chinese Academy of Sciences (CAS, Wuhan, Hubei, China). The *Anabaena pseudoichthyoides* were expanded and cultured in Blue-Green Medium (BG11) to generate enough quantities for the experiment. The culture condition of *Anabaena pseudoichthyoides* was 25 °C, 2000 Lux, with a light/dark ratio of 12 h/12 h.

After the enhanced cultivation of algae, 10 mL of *Anabaena pseudoichthyoides* was centrifuged for 10 min at 5000 r/min. Then, the supernatant in the tube was removed, and only the algal cells at the bottom were retained. To remove residual impurities, the remaining algal cells were washed with distilled water and then centrifuged and repeated three times. The leachates with different concentrations were added together with the cultivation solution to the culture flask, and the inoculation density of *Anabaena pseudoichthyoides* was about 0.002.

### 2.4. Determination of Chlorophyll a

*Anabaena pseudoichthyoides* grew after 18 days, and 10 mL of the culture solution samples were collected in tubes and centrifuged at 4500 r·min^−1^ for 10 min; the supernatant was discarded, and an equivalent of 90% methanol was added into tubes. The homogenously mixed solution was then stored at 4 °C in the dark. After 24 h extraction, the solutions were centrifuged at 4500 r·min^−1^ for 10 min, and the supernatant was collected for the determination of absorbance at 665 nm (A_665_) with a UV spectrophotometer (Lambda 350 V vis, PerkinElmer, Singapore). The chlorophyll content was then determined using the following equation:Chlorophyll a content (µg·mL^−1^) = 13.9 × A_665_(1)

### 2.5. Data Analysis

Data were analyzed using SPSS 20.0 (SPSS Inc., Chicago, IL, USA) by one-way analysis of variance (ANOVA). Origin 2021 (OriginLab Corporation, Northampton, MA, USA) was used for graphical work performance.

## 3. Results

### 3.1. The Organic Carbon Content Varied in Different Volcanic Ash Samples

Organic matter abundance is not only an important symbol for evaluating black shale but also one of the fundamental bases for evaluating the hydrocarbon potential of hydrocarbon source rocks. This result indicated that the TOC contents at the bottom of the shale layer are 1.93~4.44% and 3.0% on average. The TOC contents at the top of the layer are 3.38~5.13% and 4.0% on average (Table 1). It indicated that the TOC contents at the bottom of the shale layer are smaller than the TOC contents at the top of the layer (Figure 2). This suggests that volcanic activity has a long-term effect on biological growth. In general, the TOC contents of the Qiliao outcrop shale are high.

### 3.2. The Growth Conditions of Anabaena Pseudoichthyoides Are Different Under Different Concentrations

To investigate the effects of volcanic ash and granite on microbial growth, a simulated experiment was conducted with volcanic ash and granite immersed in water (Figure 3). There are four growth stages of *Anabaena pseudoichthyoides*: acclimatization, blooming, decline, and stabilization. By the observation of the experimental group, it is found that when microorganisms are divided into four stages, first, *Anabaena pseudoichthyoides* have a growth acclimatization period with relatively slow growth, which lasted for about 10 days. Second, they start to flourish from the 11th to the 13th day. Third, they entered the decline period around the 14th day and gradually tended to stabilize after the 15th~18th day. In contrast, in the control group, it was found that when microorganisms are divided into four stages, *Anabaena pseudoichthyoides* first have a growth acclimatization period with relatively slow growth, which lasts for about 8 days. Second, they start to flourish during the 9th to 10th day. Third, they entered the decline period around the 11th~15th day and gradually tended to stabilize around the 16th~18th day. The best microbial growth condition was observed at the 1/10 concentration of soaked volcanic ash leachate (Figure 3). This indicated that volcanic ash leachate could accelerate microbial flourishing in the early stage at 1:100 under conditions with the concentration of soaked volcanic ash leachate; the growth of the microorganisms in the experimental group was generally better than that in the control group. By observing the growth curve, we can see that the OD_665_ value at a concentration of 1:100 is greater than that of the control group on days 9 and 10. Although the difference in OD_665_ values is not significant, after 18 days of accumulation, the chlorophyll a content in the experimental group is approximately twice that of the control group. Under the 500 to 1:2500 conditions, the filtrate had almost no effect on biological growth. This is because the amount of filtrate was small and dilutes the nutrient content of the culture medium.

### 3.3. The Chlorophyll a Content Varied Under Different Concentrations

When volcanic ash leachate was combined into the culture medium with a concentration of 1/10 to 1/2500, the chlorophyll a content of *Anabaena pseudoichthyoides* was gradually decreased as the soaked volcanic ash leachate decreased, which indicated that the higher concentration of the leachate combined into the culture medium was more effective to promote the synthesis of chlorophyll a by *Anabaena pseudoichthyoides*. The results also indicated that the low concentration of volcanic ash leachate effect on the chlorophyll a synthesis of *Anabaena pseudoichthyoides* is not significant. At concentrations of 1:10 and 1:100, the experimental group of *Anabaena pseudoichthyoides* showed a significantly higher chlorophyll a content compared to the control group. Particularly at the concentration of 1:10, the chlorophyll a content in *Anabaena pseudoichthyoides* was 2~3 times greater than that of the control group. This clearly indicates that high concentrations of volcanic ash can effectively promote the growth of *Anabaena pseudoichthyoides* and enhance the synthesis of chlorophyll a (Figure 4A). The different proportions of the granite soaking filtrate resulted in little difference in the growth of the chlorophyll a produced by algae, which may be due to the low nutrient content in the granite soaking solution. Additionally, the growth conditions of the microorganisms under different volcanic ash leachate concentrations verified the same trend as the chlorophyll a content (Figure 5).

### 3.4. Mechanisms of the Effect of Nutrients on Microbial Production

Many studies have shown that nutrients promote carbohydrate and chlorophyll synthesis [21,22,23]. From Table 2, it shows that the main elemental contents of Ca^2+^, Mg^2+^, Na^+^, and K^+^ decreased by 3.8~87.24%, 75.96~92.70%, 86.56~95.67%, and 5.42~20.52% in the solution after microbial growth, respectively. The trace elements B, Ba, Zn, and Fe decreased by 27.54~94.39%, 20~82.03%, 70.45~98.29%, and 99%. This indicates that B, Ba, Zn, and Fe have been absorbed by microorganisms or participated in metabolism. At the same time, it shows that the nutrient elements in the filtrate of volcanic ash are tens of times higher than those in the filtrate of granite. This indicates that the nutrient content brought by the weathering of granite on the surface of the earth’s crust is much smaller than that brought by the dissolution of volcanic ash. Therefore, nutrients to be released from the periodic eruption of volcanic ash promote microbial production.

The statistical results of the field profile indicated that the thickness of the general volcanic ash is 2~5 cm. The calculation of the volume of each volcanic eruption is about 8 × 10^10^~20 × 10^10^ m^3^ based on the area of distribution of the high abundance of organic shale of the Longmaxi Formation. Under this circumstance, the amount of volcanic material provided by the eruption is high (the amount of volcanic ash dispersed onto the land is not included). Normally, the thickness of the sedimentary shale is 5~10 cm after each volcanic eruption, and the organisms have been flourishing during this period. Since the continuation of the Long1 section of the Longmaxi Formation is about 0.6 million years, with a sedimentary thickness of 2~3 m, the interval between the eruptions is around 200,000~400,000 years. The calculation results concluded that the time limit of volcanic ash impact could last for tens to hundreds of thousands of years. The continuous release of nutrients from volcanic ash during this period explains why microorganisms thrive and how those nutrients replenish microorganisms constantly.

## 4. Conclusions


(1)These results show that the total organic carbon (TOC) contents of these samples were distributed in the range of 1.93~5.13%, with an average content of 3.74%. In general, the TOC contents of the Qiliao outcrop shale are high. The experimental results demonstrated that the TOC contents at the bottom of the shale layer are smaller than that at the top of the layer, suggesting that the volcanic activity posed a long-term effect on biological growth.(2)The volcanic ash leachate was combined into the culture medium with a concentration of 1/10 to 1/2500, and the chlorophyll a content of *Anabaena pseudoichthyoides* was gradually decreased as the soaked volcanic ash leachate decreased, which indicated that the higher concentration of the leachate combined into the culture medium was more effective to promote the synthesis of chlorophyll a by *Anabaena pseudoichthyoides*. The results also indicated that the low concentration of volcanic ash leachate effect on the chlorophyll a synthesis of *Anabaena pseudoichthyoides* is not significant. Among all the samples, the chlorophyll a content at the 1/10 concentration was the highest, indicating that volcanic ash has the most significant effect on the promotion of chlorophyll a synthesis at a higher concentration.(3)The results show that the main elemental contents of Ca^2+^, Mg^2+^, Na^+^, and K^+^ decreased by 3.8~87.24%, 75.96~92.70%, 86.56~95.67%, and 5.42~20.52% in the solution after microbial growth, respectively. The trace elements B, Ba, Zn, and Fe decreased by 27.54~94.39%, 20~82.03%, 70.45~98.29%, and 99%. This indicates that B, Ba, Zn, and Fe have been absorbed by microorganisms or participated in metabolism.(4)The results concluded that the time limit of volcanic ash impact could last tens to hundreds of thousands of years. The continuous release of nutrients from volcanic ash during this period explains why microorganisms thriv and how those nutrients replenish microorganisms constantly.


On the basis of this study, if the dating technology meets the current requirements, it can more accurately study the duration and interval of volcanoes, as well as accurately predict the amount of eruptive material, which will have a significant impact on this research. With the advancement of technology, further research and exploration in this area are needed.

## Figures and Tables

**Figure 1 biology-13-00895-f001:**
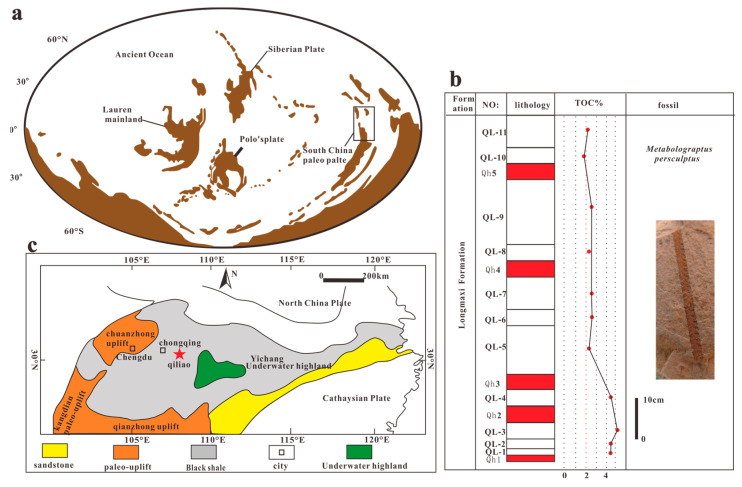
Studied region and lithological column. (**a**): Paleogeographic map of the Late Ordovician world; (**b**): Lithological columnar and content map;(QL: Shale layers, Qh: Volcanic ash layer); (**c**): Paleogeographic map and sampling location of the study area.

**Figure 2 biology-13-00895-f002:**
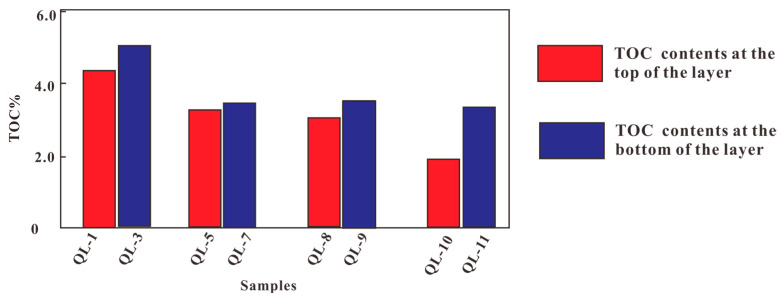
The total carbon contents in different samples.

**Figure 3 biology-13-00895-f003:**
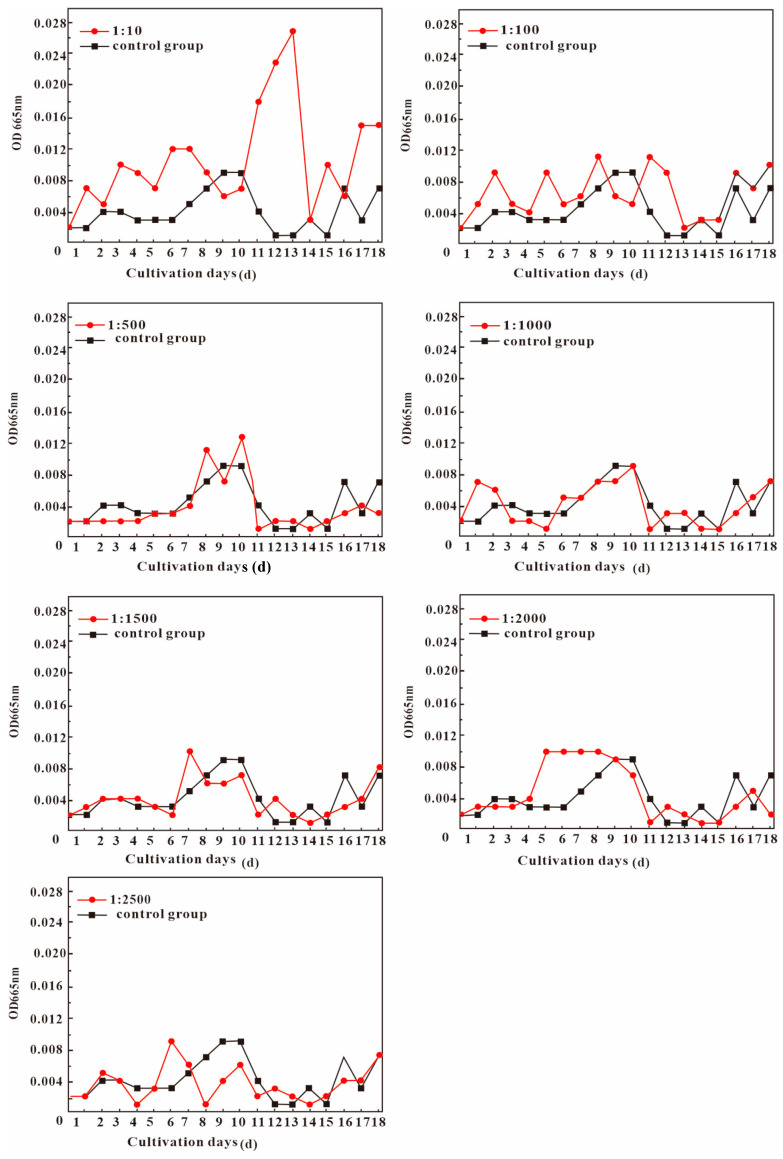
Growth curves of *Anabaena pseudoichthyoides* under different leachate concentrations.

**Figure 4 biology-13-00895-f004:**
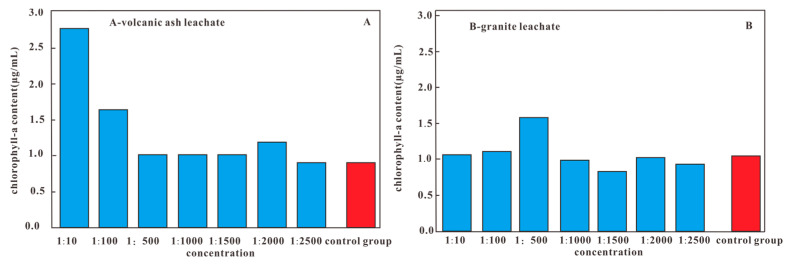
Chlorophyll a content of algae under different concentrations of volcanic ash and granite leachate. (**A**): Volcanic ash leachate; (**B**): Granite leachate.

**Figure 5 biology-13-00895-f005:**
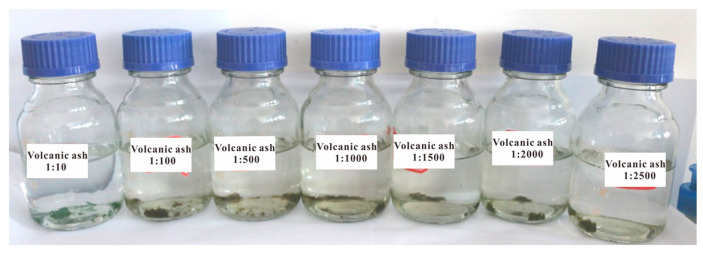
Growth of *Anabaena pseudoichthyoides* under different leachate concentrations of volcanic ash.

**Table 1 biology-13-00895-t001:** The total organic carbon (TOC) contents in the LM1 section of the Qiliao outcrop.

Code	Thickness (m)	Type	TOC (%)
QL-11	0.95–1.05	shale	3.38
QL-10	0.85–0.90	shale	1.93
QH5	0.08–0.085	volcanic ash layer	/
QL-9	0.60–0.80	shale	3.52
QL-8	0.55–0.60	shale	3.15
QH4	volcanic ash layer	volcanic ash layer	/
QL-7	0.40–0.50	shale	3.45
QL-6	0.35–0.40	shale	3.55
QL-5	0.20–0.35	shale	3.28
QH3	volcanic ash layer	volcanic ash layer	/
QL-4	0.10–0.15	shale	4.61
QH2	volcanic ash layer	volcanic ash layer	/
QL-3	0.05–0.10	shale	5.13
QL-2	0.02–0.05	shale	4.48
QL-1	0–0.02	shale	4.44
	volcanic ash layer	volcanic ash layer	/

**Table 2 biology-13-00895-t002:** Comparison of the composition of the preculture solution with that of the postculture solution.

		Filtrate of Volcanic Ash			Filtrate of Granite		
Elements		Preculture Soltionth = 1:10	Postculture (mg/L)	Reduction Rate (%)	Preculture Solution = 1:10	Postculture (mg/L)	Reduction Rate (%)
Main elements	Ca^2+^	502.5	64.12	87.24	10.9	10.486	3.80
	Mg^2+^	600.9	75.96	87.36	12	0.876	92.70
	Na^+^	1140.3	153.26	86.56	312	13.496	95.67
	K^+^	37.94	35.889	5.42	16.8	13.353	20.52
Trace elements	B	3.145	2.279	27.54	0.767	0.043	94.39
	Ba	0.128	0.023	82.03	0.003	0.0024	20.00
	Mo	<0.001	0.087	/	0.153	<0.001	
	Cd	0.692	0.295	57.37	0.0002	0.0002	0.00
	Zn	2.519	0.043	98.29	0.044	0.013	70.45
	Cu	<0.001	<0.001	/	0.04	0.03	25.00
	Fe	<0.001	<0.001	/	0.1	<0.001	
	Li	0.021	0.004	80.95	0.0002	0.0002	0.00
	Mn	2.086	0.071	96.60	0.369	0.03	91.87

## Data Availability

The data are available from the corresponding authors.

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
