# Peer review of "The Impacts of Volcanic Activity on Microbial Growth—A Simulation Experiment in the Qiliao Section in Shizhu County, Chongqing, China"

_biology, 2024, doi:10.3390/biology13110895_

Round 1

Reviewer 1 Report

Comments and Suggestions for Authors

Dears Authors I have included coments on the manuscrit, i beleaved more stadistical analysis needs to be performed among treatments to determined if obseved differences are significative or not.  

Author Response

Comment 1: It was found that B, Ba, Fe, and Zn elements decreased a lot. It will be better to indicate percentage values of decreased.

Response: Thanks for your valuable suggestion. We have made modifications to this section, introducing the percentage reduction of each element. These changes are reflected in the abstract, section 3.4, and Table 2.

Comment 2: The dissolved volcanic ash in marine water had been impacted on marine life for long time. This phrase needs to be improved

Response: Thanks for your question. We have revised this section in the second paragraph of the introduction to redescribe the impact of volcanic activity on biota.

Comment 3: The text (L 61) states that especially It is confirmed the frequent volcanic activity at P. pacificus zone of the Late Kaitian stage and the A. ascensus zone of the Early Rudan stage in the Qiliao outcrop. "It" should be "it".

Response: Thanks for your suggestion. We have made a modification in line 70 of the introduction, changing "It" to "it".

Comment 4: Fig. 1: The text (L 64) states that there is a lack of strong evidence in geologicl history that the volcanic activity impact surface productivity in the ocean currently. "geologicl" should be "geological".

Response: Thanks for your kindly and valuable suggestion. We have made a modification in line 72 of the introduction, correcting the spelling of the word.

Comment 5: In the sentence(L 143) "It indicated that volcanic ash leachate could accelerate microbial flourishing in the early stage. 1/100 under conditions with concentration of soaked volcanic ash leachate," the period (.) should be changed to a comma (,).

Response: We are sorry for our negligence, and We have made a modification in line L154.

Comment 6: The text (L 145) states that was generally a few better than that in the control group. Please indicate if results have significative differences or not.

Response: Thanks for your kindly and valuable suggestion. We have made modifications to this section in part 3.3, where we describe the growth conditions at a volcanic ash concentration of 1:100 compared to the control group.

Comment 7:The text (L 188) states that Under this circumstance, the amount of volcanic material 187

provided by the eruption is extremely high (the amount of volcanic ash dispersed onto the land is not included). May be used high or compared to other global events volcanic eruption are the ones that more materials provides.

Response: Thanks for your valuable suggestion and question. Regarding this issue, we have given it thorough thought and consideration. Due to regional and research limitations, we lack samples and data from other areas, which prevents us from conducting a multi-regional study. In future research, we will strengthen our efforts in this area.

Comment 8:In the sentence(L 205) " Through the analysis of major and trace elements in the solution before and after cultivation, it was found that B, Ba, Fe, and Zn elements decreased a lot." the period (decreased a lot.) should be changed to a comma (decreased significatively).

Response: Thanks for your kindly and valuable suggestion., and We have made a modification in line 32 of the abstract.

Comment 9:Figure 4. There are significative differences among samples

Response: Thanks for your valuable suggestion and question. We have provided some explanations for Figure 4 in lines 175-179 of section 3.3.

Reviewer 2 Report

Comments and Suggestions for Authors

The manuscript describes about the impacts of volcanic activity on microbial growth in one of the county in China. The structure of the abstract needs improvement. There is no introduction to the abstract and it contains a lot of results. The introduction needs to be re-written too. There could be future potential or future study needed at the end of the introduction. There are some grammatical errors that needs correction. The figures need editing and the legends below the figures need to be descriptive. The conclusion could have been more descriptive with the addition of ideas that you have for future studies that could help build on what you found. What challenges did the authors face with the methods that were used in this study, and how could new technology help in the future? Do you think your findings could be the same in other volcanic areas? Overall, interesting to read but revisions are required. 

Author Response

Comment: The manuscript describes about the impacts of volcanic activity on microbial growth in one of the county in China. The structure of the abstract needs improvement. There is no introduction to the abstract and it contains a lot of results. The introduction needs to be re-written too. There could be future potential or future study needed at the end of the introduction. There are some grammatical errors that needs correction. The figures need editing and the legends below the figures need to be descriptive. The conclusion could have been more descriptive with the addition of ideas that you have for future studies that could help build on what you found. What challenges did the authors face with the methods that were used in this study, and how could new technology help in the future? Do you think your findings could be the same in other volcanic areas? Overall, interesting to read but revisions are required.

Response: Thanks for your valuable suggestion and question. We have made revisions to address the issues found in the abstract, and we have added the significance of the study in the introduction. We carefully checked and thoroughly revised the grammatical aspects. Figure 2 and Table 2 have been re-edited. The conclusion has also been reorganized and modified to include the challenges faced during the research and ideas for future technologies.

Round 2

Reviewer 2 Report

Comments and Suggestions for Authors

The corrections are made as per my suggestions. Thanks!